# A Nanoengineered Stainless Steel Surface to Combat Bacterial Attachment and Biofilm Formation

**DOI:** 10.3390/foods9111518

**Published:** 2020-10-22

**Authors:** Ga-Hee Ban, Yong Li, Marisa M. Wall, Soojin Jun

**Affiliations:** 1Department of Human Nutrition, Food, and Animal Sciences, University of Hawaii, 1955 East-West Road, Honolulu, HI 96822, USA; gaheeban87@gmail.com (G.-H.B.); liyong@hawaii.edu (Y.L.); 2Daniel K. Inouye U.S. Pacific Basin Agricultural Research Center, 64 Nowelo Street, Hilo, HI 96720, USA; marisa.wall@ars.usda.gov

**Keywords:** electrochemical etching, nanotechnology, stainless steel, antibiofilm, *Escherichia coli* O157:H7, *Salmonella* Typhimurium

## Abstract

Nanopatterning and anti-biofilm characterization of self-cleanable surfaces on stainless steel substrates were demonstrated in the current study. Electrochemical etching in diluted aqua regia solution consisting of 3.6% hydrogen chloride and 1.2% nitric acid was conducted at 10 V for 5, 10, and 15 min to fabricate nanoporous structures on the stainless steel. Variations in the etching rates and surface morphologic characteristics were caused by differences in treatment durations; the specimens treated at 10 V for 10 min showed that the nanoscale pores are needed to enhance the self-cleanability. Under static and realistic flow environments, the populations of *Escherichia coli* O157:H7 and *Salmonella* Typhimurium on the developed features were significantly reduced by 2.1–3.0 log colony-forming unit (CFU)/cm^2^ as compared to bare stainless steel (*p* < 0.05). The successful fabrication of electrochemically etched stainless steel surfaces with Teflon coating could be useful in the food industry and biomedical fields to hinder biofilm formation in order to improve food safety.

## 1. Introduction

Biofilms are sturdy microbial communities commonly found at liquid–air interfaces, and are the cause of many medical, industrial, and environmental problems [1,2]. Within these biofilms, bacteria attach to, multiply, and encase themselves in a slimy surface-associated community of extracellular polymeric substances (EPS). The EPS is generally composed of proteins, lipids, polysaccharides, and nucleic acids [3]. Importantly, bacterial cells embedded in this EPS layer tend to express phenotypic features that differ from those of their planktonic counterparts, such as augmented tolerance to antibacterial agents [1].

Two bacterial pathogens, *Escherichia coli* O157:H7 and *Salmonella enterica*, have been shown to form biofilms on solid surfaces, including plastic, metal, glass, and rubber [4,5,6,7]. The presence of these biofilms is an ongoing concern in the food industry, as it has been demonstrated that this may lead to enhanced environmental persistence [8,9]. In humans, *E. coli* O157:H7 is a major cause of various illnesses, including bloody diarrhea and hemolytic uremic syndrome [10]. Additionally, *S*. Typhimurium infection leads to fever, chills, diarrhea, nausea, vomiting, and abdominal pain [11]. Biofilm contamination within both the food processing and biomedical sectors is a common and significant source of human infections, outbreaks, and recalls [12,13].

Surface technology has been investigated as a means of inhibiting bacterial adhesion and biofilm development [14,15,16,17]. This approach typically relies on the release of biocidal agents or inhibition of bacterial adhesion [18]. For instance, such strategies include coatings that release microbiocidal agents, such as silver, antibiotics, enzymes, polycations, and antimicrobial peptides, into the surrounding aqueous environment [16]. On the other hand, adhesion prevention strategies have been used to assess surface chemical functional groups that hinder protein adsorption, such as weakly polarizable materials that reduce van der Waals forces [19], low surface energy [20], and hydrophilic polymeric materials that create highly hydrated features [21]. However, these approaches are mostly transient, and it is difficult to combat bacterial attachment by surface structuring or surface chemistry alone [22]. As a result, slippery liquid-infused porous surfaces (SLIPS) were proposed for anti-biofouling [22,23]. In 2016, however, the United States Food and Drug Administration (FDA) revoked the use of perfluorinated chemicals, which are used for SLIPS involving food contact [24].

Nanostructures, which can effectively prevent attachment of water droplets, were initially observed on the lower surface of a lotus leaf [25]. Leaves of the lotus flower are extremely hydrophobic and are thought to have “self-cleaning” characteristics, as water droplets completely roll over the surface, removing undesirable particles [26]. Furthermore, several researchers have observed nanoscale features on softwood fiber, titanium, and aluminum that prevent bacterial adhesion (i.e., smaller than the size of a bacterium) [27,28,29].

Stainless steel is an alloy material that is widely used in the food industry and in various environments because of its superior corrosion resistance and workability [30]. Currently, there are very limited nanoengineered surface studies that have assessed the combination of etching techniques and Teflon coating and their inhibitory effects for bacterial adhesion on stainless steel. Etching, the simplest method used to fabricate a nanostructured surface, creates the nanofeatures on the surface arbitrarily, however the overall roughness is homogeneous [31].

Previous studies have shown the surface topography and surface chemical composition to strongly affect cell behavior, including bacterial attachment and biofilm development. Therefore, the main purpose of this study was to investigate the effects of a Teflon-coated electrochemically etched surface (ET) on the adhesion capabilities of *E. coli* O157:H7 and *S.* Typhimurium, and to evaluate their biofilm formation ability under static and dynamic conditions.

## 2. Materials and Methods

### 2.1. Fabrication of Nanoengineered Stainless Steel Surface Using Electrochemical Etching

The 304 stainless steel was cut into 50 × 25 × 0.2 mm specimens to serve as the substrate for the nanoengineered surfaces. Each specimen was submerged in a diluted aqua regia solution, which was mingled in a 1:1 ratio (*v:v*) of 3.6% (1N) HCl and 1.2% HNO_3_ and placed in parallel to a carbon plate at a distance of 5 cm. Both were positioned in a double-walled beaker set at 4 °C and connected to a circulation temperature controller (Figure 1). Constant electric potentials at 5, 10, and 15 V provided by a DC power supply (CPX400SP; Aim TTI, Huntingdon, United Kingdom) were applied to the stainless steel specimen for 5, 10, and 15 min. The stainless steel was employed as a working electrode (anode) and the carbon was used as a counter electrode (cathode) in the electrochemical etching. The substrates were rinsed in deionized (DI) water immediately after etching and completely dried. This method was adapted from Lee et al. [32] with minor modifications.

### 2.2. Creation of Superhydrophobic Surface on Stainless Steel

For the superhydrophobic structure, the electrochemically etched surfaces (E) were coated with 0.2 wt % Teflon AF1600 powder (DuPont Inc., Wilmington, DE, USA) in FC-40 perfluoro compound (Sigma-Aldrich, St. Louis, MO, USA). Each specimen was consecutively baked on a hot plate at 110 °C for 10 min, at 165 °C for 5 min, and at 330 °C for 15 min. Finally, the samples were rinsed with DI water for 5 min and air-dried before water contact measurements and bacteriological tests. The measurements of apparent contact angles of sessile water droplets (~3 μL) were conducted on each specimen using a contact angle goniometer (FTA-1000, First Ten Angstroms, Newark, CA, USA).

### 2.3. Bacterial Strains and Culture Preparation

*E. coli* O157:H7 C7927 and *S*. Typhimurium ATCC 14028 were provided by the Food Microbiology Laboratory (University of Hawaii at Manoa, Honolulu, HI). Each strain of *E. coli* O157:H7 and *S*. Typhimurium was grown in 10 mL of tryptic soy broth (TSB) at 37 °C for 24 h under static incubating conditions, collected by centrifugation at 4000× *g* at 4 °C for 20 min, and washed three times with 100 mM phosphate-buffered saline (PBS) at pH 7.1–7.4. The final pellets were resuspended in sterile PBS corresponding to approximately 10^8^–10^9^ colony-forming unit (CFU)/mL.

### 2.4. Bacterial Attachment and Biofilm Forming Ability Test

For cell attachment, we adapted a method from Kim et al. [33]. Each fabricated specimen was moved to a sterile 50 mL centrifuge tube (VWR International, Radnor, PA, USA) containing 10 mL bacterial cell suspensions in PBS (≈10^8^ CFU/mL). The tubes containing the specimens were kept at 4 °C for 24 h for cell attachment. The cell-attached specimens were transferred and submerged in 200 mL of sterile distilled water (22 ± 2 °C) and softly agitated for 5 s. Rinsed samples were placed in 50 mL tubes containing 30 mL of TSB, then stored at 25 °C for 6 days of static incubation.

Biofilms were generated after bacterial attachment according to the methodology previously described by Kim et al. [34] based on the Center for Disease Control (CDC) biofilm reactor, which contained eight specimens (Biosurface Technologies, Corp., Bozeman, UT, USA). Bacterial cultures were prepared by incubating *E. coli* O157:H7 and *S.* Typhimurium in 1/10 strength TSB at 37 °C for 20 h. The reactor was placed on a stirring plate connected to the peristaltic pump (Manostat Corp., New York, NY, USA). Then, 3.5 mL of the bacterial culture solution was inoculated into 350 mL of 1/100 strength TSB and added to the biofilm reactor. The initial bacterial population in this medium was approximately 10^6^ CFU/mL. The developed nanoengineered coupons were placed in the reactor, which was operated in batch mode for 24 h at 100 rpm at room temperature. After completion of the batch mode, the reactor was then connected to a carboy containing 1/300 strength TSB and a waste collection carboy. The system was then operated in continuous flow mode at a rate of 11.67 mL/min for 24 h. Differential strengths of TSB were applied to the batch and continuous flow modes to maintain the final volume and concentration.

### 2.5. Bacterial Enumeration

After the attachment process, the specimens were removed and placed in 50 mL tubes containing 10 mL of PBS and 2 g of sterile glass beads (425–600 μm; Sigma-Aldrich, St. Louis, MO, USA), then vortexed at maximum speed for 1 min. Subsequently, cell suspensions were serially diluted ten-fold in 0.1% pepton water (Difco, Sparks, MD, USA), then 0.1 mL of undiluted cell suspension or dilution was spread onto sorbitol–MacConkey agar (SMAC; Difco, Sparks, MD, USA) and xylose lysine desoxycholate agar (XLD; Difco, Sparks, MD, USA) to enumerate *E. coli* O157:H7 and *S*. Typhimurium, respectively.

### 2.6. Field Emission Scanning Electron Microscopy (FE-SEM)

In order to characterize the nanoscale features of the fabricated surfaces and visually ascertain the effects of electrochemical etching and Teflon coating on bacterial adhesion, the developed surfaces were photographed using field emission scanning electron microscopy (FE-SEM, Hitachi S-4800, Ibraraki, Japan). The specimens were submerged in 2.5% glutaraldehyde suspended in 0.1 M sodium cacodylate buffer (pH 7.4) at 25 °C for 1 h and washed in 0.1 M cacodylate buffer twice for 10 min each. For post-fixation, specimens were submerged in a mixture of 1% osmium tetroxide and 0.1 M cacodylate buffer for 30 min. The specimens were dried with a graded ethanol series of 10, 20, 30, 50, 70, 85, and 95% (5 and 15 min incubation times for each change), and then three changes of 100% for 10 min each. After dehydration, the specimens were positioned in a critical point dryer filled with liquid carbon dioxide and softly dehydrated by evaporating liquid carbon dioxide. Dried samples were mounted onto aluminum stubs using conductive carbon tape and coated with gold–palladium in a Hummer 6.2 sputter coater using a vacuum coater. Finally, photomicrographs were obtained using FE-SEM.

### 2.7. Statistical Analysis

All experiments were repeated at least three times with independently prepared samples. All bacterial counts of the specimens were log-transformed. The data were analyzed via analysis of variance and Duncan’s multiple range tests (the separation of means was tested at a probability level of *p* < 0.05) using the Statistical Analysis System (SAS Institute, Cary, NC, USA).

## 3. Results and Discussion

### 3.1. Surface Morphology of Nanoengineered Stainless Steel Surface

We observed a morphology typical of a stainless steel surface (e.g., possessing deep crevices) (shown in Figure 2a,b). Crevasses within stainless steel are difficult to clean and represent protective harborages for bacterial cells [35]. In contrast, Figure 2c,d show the FE-SEM images of nanostructures developed on the 304 stainless steel substrates using the electrochemical etching technique. The electrochemical etching conditions, including the applied voltage and treatment time, were optimized for the desired nanostructure formation via parameter optimization. As a result, the morphology of the developed E treated at 10 V for 10 min had distinct nanoscale pores. This is in accordance with Lee et al. [32], who observed that a substrate etched at 10 V had nanopores and microsized bumps that resulted in a hierarchical surface and a significant change in wettability. While many researchers have fabricated well-ordered nanoporous aluminum and nanopillared structures by transforming the hexagonal array of nanoporous aluminum [28,36], stainless steel is a steel alloy, meaning formation of uniform surface nanostructures is difficult. Song et al. [37] previously observed that after anodization on pure titanium (grades 2 and 3) and titanium alloys (Ti6Al4V and TiAl7Nb), nanopores were distributed regularly over the surface of anodically oxidized pure titanium metals, however the distribution and size of the nanopores on the anodically oxidized titanium alloy varied. Based on the FE-SEM micrographs, the pore sizes of the nanopore structures on E were between 50 and 100 nm. The adhesion of Staphylococci bacteria, which are approximately 1 μm in diameter and spherical, dramatically decreased on poly(ethylene glycol) when the distance between the microgels was 1.5 μm or less [38]. This finding implies that the surface structures for operational inhibition of bacterial deposition should be nanosized, which is smaller than the dimensions of a bacterium.

### 3.2. Wetting Properties of Nanoengineered Stainless Steel Surface

The water contact angles of the resulting electrochemical etching and Teflon coating were measured, with the results shown in Figure 3. The contact angle on the flat bare stainless steel surface before electrochemical etching was 89.5° (Figure 3a), indicating low hydrophobicity. Nanoscale features on a solid surface are essential for superhydrophobicity, which can generate a high contact angle. The contact angle on the etched surface without Teflon coating was 128.2° (Figure 3b). Lee et al., (2015) reported that E was hydrophobic, but the water droplet adhered onto the E instead of rolling over it, which is characteristic of hydrophobicity and illustrates the rose petal effect related to high contact angle hysteresis [39]. On the other hand, the contact angle measured on the E when coated with Teflon increased to 151.1° (Figure 3c). Accordingly, the Teflon coating influenced the surface superhydrophobicity, having a water contact angle greater than 150°.

When tilted, a water droplet rolled off the ET (Figure 4b,c), as compared to the bare stainless steel, which held the water droplet due to capillary interactions (i.e., surface tension) between the surface and the liquid (Figure 4a). When a water droplet contacts a rough surface possessing an appropriate combination of surface texture and solid–liquid surface energy in such a configuration, water may not fully enter into the surface pores, but could rather position itself on the posts and form a composite solid–liquid–air interface, in line with the Cassie–Baxter model [40,41]. The Cassie–Baxter state brings about liquid-repellent surfaces, and the high liquid–air fractional contact area facilitates self-cleaning [42]. Such self-cleaning, nanostructured, superhydrophobic features can be useful in diminishing bacterial interactions with surfaces.

### 3.3. Reduced Bacterial Adhesion and Biofilm Formation on a Nanoengineered Stainless Steel Surface

Figure 5 displays the attachment responses of *E. coli* O157:H7 and *S*. Typhimurium on bare stainless steel; E treated at 10 V for 5, 10, and 15 min; and E treated at 10 V for 5, 10, and 15 min and followed by Teflon coating. The initial levels of *E. coli* O157:H7 and *S*. Typhimurium in PBS were 7.9 and 8.2 log CFU/cm^2^, respectively; and the attached populations of these bacteria on the control stainless steel measured 6.6 and 6.7 log CFU/cm^2^, respectively. The attachment levels of *E. coli* O157:H7 and *S*. Typhimurium were reduced by 1.1–1.5 log CFU/cm^2^ and 0.8–1.5 log CFU/cm^2^ on the E treated at 10 V, respectively. Slight reductions of bacterial attachment occurred on the E because of the diminished contact angle and the resulting water deposition led to bacterial adhesion. Enhancing the contact angle from 128° to 151° on the ET reduced attachment levels of *E. coli* O157:H7 and *S*. Typhimurium by 1.6–2.3 log CFU/cm^2^ and 2.5 log CFU/cm^2^, respectively. This was caused by the weakly polarizable Teflon layer with low surface energy on the nanoporous feature, minimizing the van der Waals interaction between the bacteria and solid surface [43]. Previously, we showed that attached *E. coli* K-12 decreased by 1.5 log CFU/cm^2^ on the Teflon-coated large nanoporous anodic aluminum oxide when compared to the bare aluminum [44]. Hizal et al. [28] reported that Teflon coating significantly decreased the adhesion levels of *Staphylococcus aureus* and *E. coli* as compared to the flat surface. In the present study, attachment levels between *E. coli* O157:H7 and *S*. Typhimurium were not significantly different (*p* > 0.05) on the developed surfaces, but a significant decrease in the population of *S*. Typhimurium was detected on the ET treated for 5 and 10 min. Fadeeva et al. [45] observed different attachment responses for *S. aureus* and *Pseudomonas aeruginosa* on superhydrophobic titanium surfaces made using femtosecond laser ablation. Although colonization of *P. aeruginosa* cells was not seen on the fabricated titanium structures, in contrast to *S. aureus*, its biovolume was comparable to that observed for *S. aureus* because *P. aeruginosa* created a substantial amount of EPS [45]. Wang et al. [46] reported that both *E. coli* O157:H7 and *S*. Typhimurium strains had different abilities to produce EPS, which can affect bacterial growth and colonization on solid surfaces.

Bacterial attachment is considered the first step of biofilm formation and biofouling, and involves the accumulation of bacterial cells and organic materials. Under static conditions, the *E. coli* O157:H7 and *S*. Typhimurium cells in biofilm formed on the ET significantly decreased by 2.4 and 2.1 log CFU/cm^2^, compared to the bacteria on the E, which decreased by 0.9 and 0.4 log CFU/cm^2^, respectively (Figure 6a). When immersed, biofilm formation and development usually occurs under dynamic flow conditions (e.g., in food processing facilities, venous catheters, and dental water lines), and biofilms have been known to attach strongly to surfaces under flow [47]. For this reason, biofilm formation was tested on the developed surfaces placed in the CDC reactor, which imitates nature-like shear forces and renewable nutrient sources. In the present study, the levels of biofilm-forming *E. coli* O157:H7 and *S*. Typhimurium cells on the ET significantly decreased by 2.9 and 3.0 log CFU/cm^2^ (*p* < 0.05), respectively, as compared to bare stainless steel. Slightly additional reductions of bacterial biofilm cells on the ET were observed under flow conditions. Fluid shear stresses can cause detachment of bacterial cells by slipping and rolling off surfaces [48]. Hizal et al. [28] observed that nanopillared surfaces with Teflon coating caused the greatest reductions under flow, with more pronounced effects in *S. aureus* than *E. coli*.

Yin et al. [49] reported that the use of SLIPS on the enamel surface significantly hindered biofilm development of *Streptococcus mutans* in vitro. In another study, SLIPS prevented 99.6% of *P. aeruginosa* biofilm adhesion over a 7 day duration, as well as 97.2% of *S. aureus* and 96% of *E. coli* biofilm attachment under static and low flow conditions [22]. Comparatively, in our study, ET reduced more than 99% of *E. coli* O157:H7 and *S*. Typhimurium in terms of attachment and biofilm development.

After 4 h of the bacterial attachment treatment, the density of *E. coli* O157:H7 attached on the bare stainless steel was higher than the developed surface (Figure 7). While dense biofilm coverage with EPS was seen on the bare stainless steel, we observed sparse biofilm cells on the ET under static conditions. Similarly, under flow conditions, clustered bacterial cells surrounded by biofilm mass were observed on the bare stainless steel, whereas poor biofilm deposition was observed on the ET. Further investigations are needed to enhance the nanostructure of stainless steel, and to understand the interactions between bacteria and the nanofeatures of stainless steel.

Electrochemical etching followed by Teflon coating treatment has been shown to be an effective strategy for hindering bacterial attachment and biofilm development on stainless steel in laboratory settings. As stainless steel is frequently used in the food industry, the developed nanoengineered surfaces could be a preventive control for microbial populations in food production sites, without requiring heat or chemical involvement. Biofilm formation of *E. coli* O157:H7 and *S*. Typhimurium on the developed surface significantly decreased in both static and flow aquatic environments as compared to bare stainless steel (*p* < 0.05). The combination of nanostructured stainless steel and coating with a low surface energy material holds great promise for antibiofilm and antibiofouling applications, including water systems, food industry settings, and biomedical spaces where bacterial adhesion is widespread. For commercialization of this technique for mass production in the food industry, the wear rate of the nanostructured stainless steel with Teflon coating and scaling factors in consideration of the etching process should be investigated in future studies.

## Figures and Tables

**Figure 1 foods-09-01518-f001:**
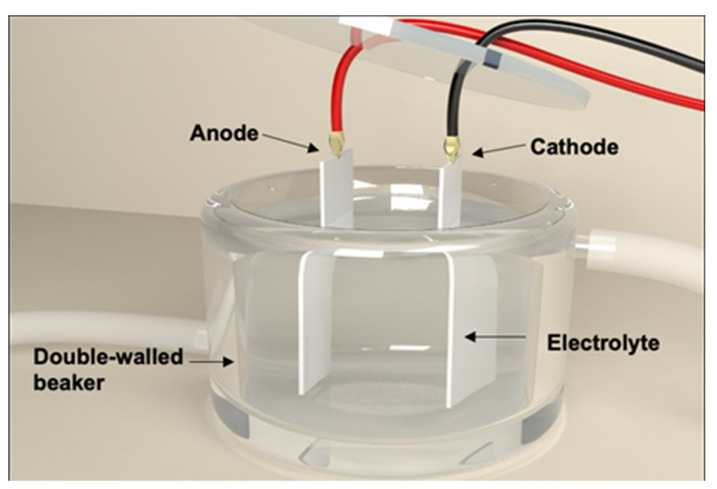
Schematic diagram of the fabrication system for the electrochemically etched stainless steel used in this study at the University of Hawaii.

**Figure 2 foods-09-01518-f002:**
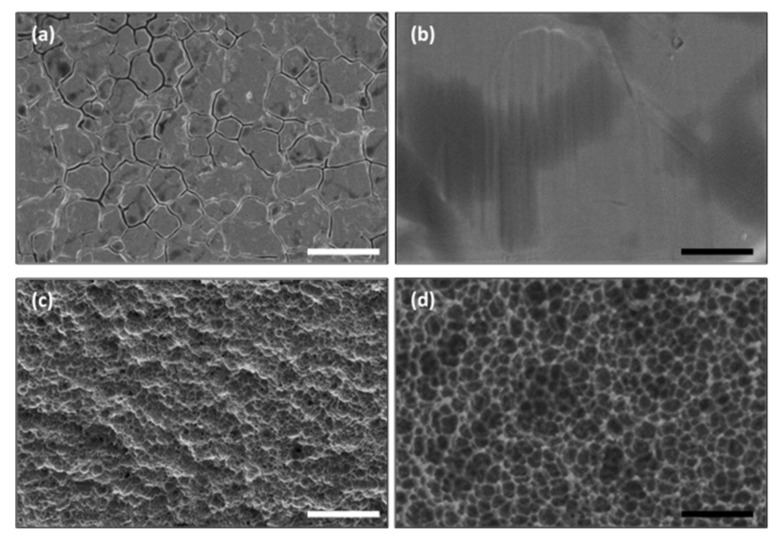
FE-SEM micrographs of bare stainless steel (**a**,**b**) and stainless steel electrochemically etched at 10 V for 10 min (**c**,**d**). White and black scale bars in (**a**–**d**) indicate 25 μm and 500 nm, respectively.

**Figure 3 foods-09-01518-f003:**
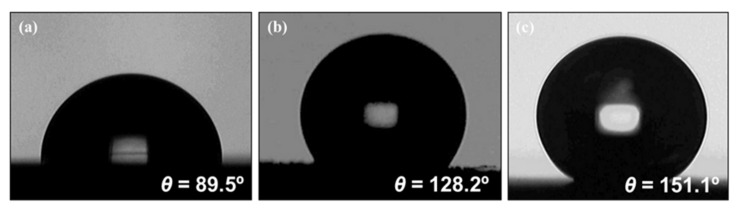
Surface wettability of (**a**) bare stainless steel, (**b**) stainless steel electrochemically etched at 10 V for 10 min, and (**c**) stainless steel electrochemically etched at 10 V for 10 min with Teflon coating.

**Figure 4 foods-09-01518-f004:**
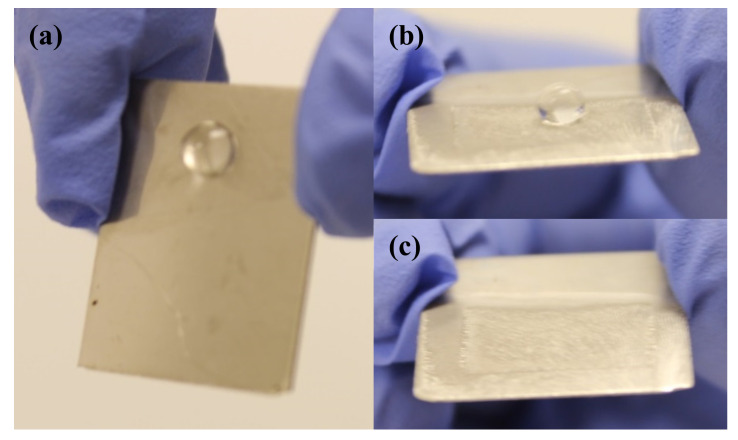
Behaviors of a water droplet on a bare stainless steel surface (**a**) and the electrochemically etched surface with Teflon coating (**b**), (**c**) when tilted.

**Figure 5 foods-09-01518-f005:**
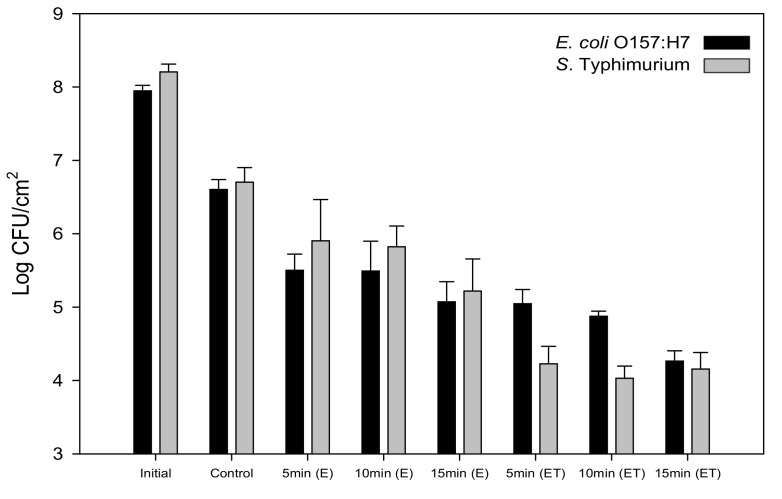
Populations (log CFU/cm^2^) of *E. coli* O157:H7 and *S*. Typhimurium attached to bare stainless steel; stainless steel electrochemically etched 10 V for 5, 10, and 15 min (E); and stainless steel electrochemically etched at 10 V for 5, 10, and 15 min with Teflon coating (ET).

**Figure 6 foods-09-01518-f006:**
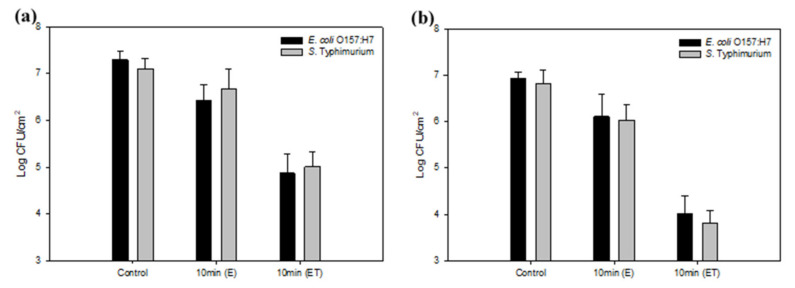
Biofilm formation of *E. coli* O157:H7 and *S*. Typhimurium in static (**a**) and dynamic (**b**) states. Populations of bacteria (log CFU/cm^2^) attached to bare stainless steel, stainless steel electrochemically etched at 10 V for 10 min (E), and stainless steel electrochemically etched at 10 V for 10 min with Teflon coating (ET).

**Figure 7 foods-09-01518-f007:**
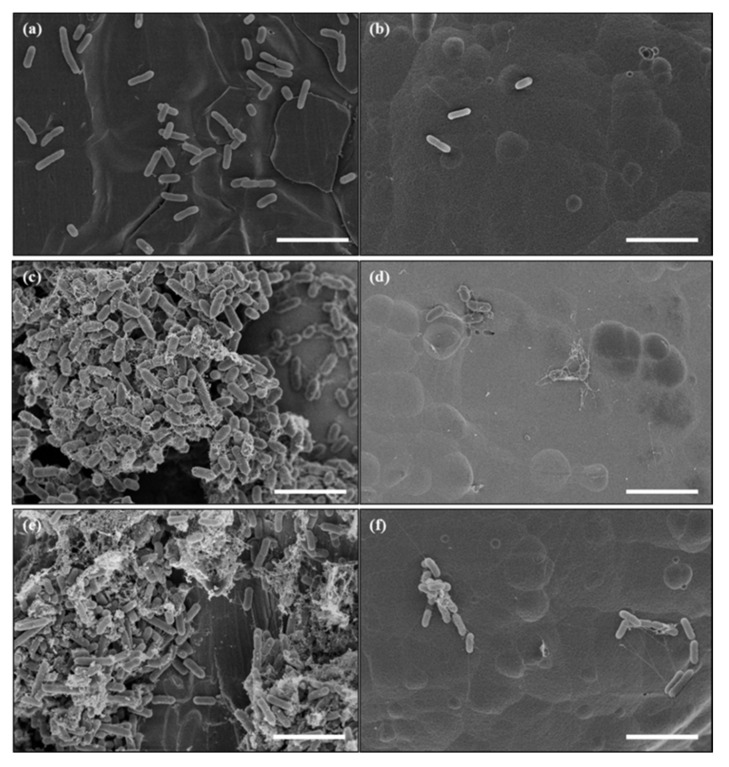
Comparison of bacterial attachment and biofilm formation of *E. coli* O157:H7 on bare stainless steel (**a**,**c**,**e**) and nanoengineered stainless steel (stainless steel electrochemically etched at 10 V for 10 min with Teflon coating) (**b**,**d**,**f**) observed by FE-SEM. Bacteria were attached on the surfaces after being submerged in bacterial cell suspension for 4 h (**a**,**b**), under static (**c**,**d**) and flow conditions (**e**,**f**). White scale bars in (**a**–**f**) indicate 5 μm.

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
