# Peer review of "A Nanoengineered Stainless Steel Surface to Combat Bacterial Attachment and Biofilm Formation"

_foods, 2020, doi:10.3390/foods9111518_

Round 1

Reviewer 1 Report

Figure 1 needs to also include images of teflon-coated surfaces, as these appear to be what was used most successfully for bacterial inhibition.

The images in Figure 7 (presumed teflon coated as surface looks very different to those shown in Figure 1 and is described in text as ET; but this is not mentioned in the legend) do not seem to be consistent with the cell count data presented in Figure 6. Figure 6 shows E. coli control and teflon coated cell counts in a static state go from ~7.3 to ~ 5 Log CFU/cm2 respectively; and in a dynamic state from ~7 to ~4 Log CFU/cm2 respectively. Yet the images show much more reduction of bacteria. With this I have to wonder how representative these images are? Alternatively, why do these data not match?

Why do the surfaces of the nanoengineered steel in Figure 7 look so different to Figure 1? If it is the teflon coating, then this appears to cover the small pores created by etching – so then what is the point of etching?

Legends, methods, and data need clarification to better understand the images and data shown.

Author Response

Figure 1 needs to also include images of teflon-coated surfaces, as these appear to be what was used most successfully for bacterial inhibition.

The authors believe that the reviewer discussed about Figure 2, not Figure 1. Figure 2 was primarily aimed to show nano-porous structures created on the surfaces by comparing FE-SEM micrographs between bare stainless steel and electrochemical etched stainless steel.

The images in Figure 7 (presumed teflon coated as surface looks very different to those shown in Figure 1 and is described in text as ET; but this is not mentioned in the legend) do not seem to be consistent with the cell count data presented in Figure 6. Figure 6 shows E. coli control and teflon coated cell counts in a static state go from ~7.3 to ~ 5 log CFU/cm2 respectively; and in a dynamic state from ~7 to ~4 log CFU/cm2 respectively. Yet the images show much more reduction of bacteria. With this I have to wonder how representative these images are? Alternatively, why do these data not match?

Figure 7 legend was revised for clarification. As the reviewer pointed out in Figure 6, the bacterial counts experienced 2 to 3 log reductions when the Surface ET replaced the bare controls either in a static or dynamic state, which indicates microbial counts disappeared or detached from the surfaces by 99% or 99.9%. We believe that this finding would be in a good agreement with FE-SEM observations although the experimenters had to include false-positive bacterial colonies which might originate from the coupon edges which was not fully electrochemically treated (Please see 2.5 Bacterial enumeration). Those false-positive M/O counts would apply for both bare controls and nano-engineered surfaces.

Why do the surfaces of the nanoengineered steel in Figure 7 look so different to Figure 1? If it is the teflon coating, then this appears to cover the small pores created by etching – so then what is the point of etching?

The authors believe that the reviewer discussed about Figure 2, not Figure 1. Figures 7 and 2 have different scale bars as indicated in the Figure captions. In general, Teflon coating was applied onto the surface roughness in a thin nanoscale layer (50 – 100 nm) using the precision control. Therefore, the coating was not intended to cover/fill the porous structures on the engineered surfaces. Also a typical water contact angle on the Teflon-coated surface (without any nano-engineered surface roughness) is 114°, which is much lower than those of Surfaces E (128°) and ET (151°). There is no way Teflon coating itself would improve the water contact angle up to 151° (the level of superhydrophobicity) without the morphological contribution of the electrochemically etched surface.

 Legends, methods, and data need clarification to better understand the images and data shown.

The relevant sentences and paragraphs were revised for further clarification.

Reviewer 2 Report

This is a well written paper dealing with nano-engineering of stainless Steel Surface to inhibit bioflim formation. Results are interesting and novel, but there are some aspects that need to be addressed.

L 86-87, the description of the superhydrophobic structure coated with teflon seems to be repeated to some extent. Please revise.

L 96, were the bacteria grown shaken or under static conditions?

L 98-99, please indicate the microbial counts correctly (10(8)-10(9) and in the rest of the text.

L 107, the paragraph is very confusing and does not clearly explain how the biofilm was produced or why different strengths of TSB were used.

L 141-145, it is not part of the results of this study. It should be moved to introduction or adapted for the discussion, once results are presented.

Results presented are very interesting, specially the reduction of microbial populations in electrochemically etched stainless Steel with Teflon coating. This study is performed in laboratory conditions, this should be mentioned further in the discussion. Also the technical feasibility to apply this technology in industrial conditions should be mentioned.

Author Response

This is a well written paper dealing with nano-engineering of stainless Steel Surface to inhibit bioflim formation. Results are interesting and novel, but there are some aspects that need to be addressed.

L 86-87, the description of the superhydrophobic structure coated with teflon seems to be repeated to some extent. Please revise.

The sentence was revised for clarification.

L 96, were the bacteria grown shaken or under static conditions?

The bacteria were grown under a static condition. We have added the sentence reading “Each strain of E. coli O157:H7 and S. Typhimurium was grown in 10 mL of TSB at 37°C for 24 h under static incubating conditions” in lines 96-99 for clarification.

L 98-99, please indicate the microbial counts correctly (10(8)-10(9) and in the rest of the text.

The wordings were revised in the whole manuscript.

L 107, the paragraph is very confusing and does not clearly explain how the biofilm was produced or why different strengths of TSB were used.

The relevant paragraph was revisited and revised for further clarification, as read:

“Biofilms were generated after bacterial attachment according to the methodology previously described by Kim et al. [34] based on the CDC biofilm reactor  which occupied eight specimens (Biosurface Technologies Inc.). Bacterial cultures were prepared by incubating E. coli O157:H7 and S. Typhimurium in 1/10-strength TSB at 37°C for 20 h. The reactor was placed on a stir plate, connected to the peristaltic pump (Manostat Corp.). A 3.5 mL of the bacterial culture solution was then inoculated into 350 mL of 1/100-strength TSB and added to the biofilm reactor. The initial bacterial population in this medium was approximately 106 CFU/mL. The developed nano-engineered coupons were placed in the reactor, which were operated in a batch mode for 24 h at 100 rpm at room temperature. After the completion of the batch mode, the reactor was then connected to a carboy containing 1/300-strength TSB and a waste collection carboy. The system was then operated at the continuous flow mode at a rate of 11.67 mL/min for 24 h. Differential strengths of TSB were applied to the batch and continuous flow modes in order to maintain the final volume and concentration.”

L 141-145, it is not part of the results of this study. It should be moved to introduction or adapted for the discussion, once results are presented.

As the reviewer suggested, the relevant paragraph was relocated in the Introduction section.

Results presented are very interesting, specially the reduction of microbial populations in electrochemically etched stainless Steel with Teflon coating. This study is performed in laboratory conditions, this should be mentioned further in the discussion. Also the technical feasibility to apply this technology in industrial conditions should be mentioned.

The authors revised the discussion part thoroughly as the reviewer pointed out (Lines 268-279).